# The Development of FAK Inhibitors: A Five-Year Update

**DOI:** 10.3390/ijms23126381

**Published:** 2022-06-07

**Authors:** Andrea Spallarossa, Bruno Tasso, Eleonora Russo, Carla Villa, Chiara Brullo

**Affiliations:** Department of Pharmacy, University of Genova, Viale Benedetto XV, 3, 16132 Genova, Italy; andrea.spallarossa@unige.it (A.S.); bruno.tasso@unige.it (B.T.); russo@difar.unige.it (E.R.); carla.villa@unige.it (C.V.)

**Keywords:** focal adhesion kinase, FAK inhibitors, anticancer compounds, medicinal chemistry, PROTAC, pyrimidines, triazines

## Abstract

Focal adhesion kinase (FAK) is a non-receptor tyrosine kinase over-expressed in different solid cancers. In recent years, FAK has been recognized as a new target for the development of antitumor agents, useful to contrast tumor development and metastasis formation. To date, studies on the role of FAK and FAK inhibitors are of great interest for both pharmaceutical companies and academia. This review is focused on compounds able to block FAK with different potencies and with different mechanisms of action, that have appeared in the literature since 2017. Furthermore, new emerging PROTAC molecules have appeared in the literature. This summary could improve knowledge of new FAK inhibitors and provide information for future investigations, in particular, from a medicinal chemistry point of view.

## 1. Introduction

Focal adhesion kinase (FAK), reported in 1991 by Guan and co-workers [1], is a non-receptor 125 kDa tyrosine kinase encoded by the human gene located on chromosome 8 [2]. In detail, it occupies the site of focal adhesion between cells and the extracellular environment, and plays an essential role in the survival of anchorage-dependent cells and in integrin-mediated cell migration [3]. Particularly, FAK receives different extracellular signals from cell-surface transmembrane receptors, such as integrins, cytokines, growth factors and G protein-coupled receptors, triggering different intracellular pathways involved in a variety of cellular activities.

FAK consists of three major domains: an N-terminal FERM homology domain, a central kinase domain and a C-terminal FAT (Focal Adhesion Targeting) domain. These three domains are separated by three proline-rich sequences (PRP1, PRP2 and PRP3) [4]. In detail, FAK protein presents six phosphorylation tyrosine kinase sites (namely, Tyr397, Tyr407, Tyr576, Tyr577, Tyr861 and Tyr925), and the FERM domain is able to interact with different receptor tyrosine kinases, such as epidermal growth factor receptor (EGFR) and platelet-derived growth factor receptor (PDGFR).

Interactions of FAK kinase domain with FERM, PRI/PR-II and FAT regions prevent Tyr397 phosphorylation, thus inducing FAK auto-inhibition. When integrin binds to the extracellular matrix, Tyr397 is phosphorylated, causing FAK activation and interaction with different intracellular pathways, such as Src, growth factor receptor-bound protein (Grb2), Mitogen-activated protein kinase (MAPK) cascade and p130cas. All these events cause focal adhesion complex formation, cell adhesion, cell migration, differentiation and, finally, cancer cell growth and progression, and, even if different studies have reported contradictory results, the events demonstrate that FAK function depends on specific cell behavior [5]. In particular, FAK activation through Src-mediated phosphorylation of Tyr576 and Tyr577 causes a protein conformational change that prevents the intramolecular interaction between the N-terminal FERM domain and the FAK-kinase domain [6].

In detail, FAK/Src complex results in:(1)Rho/Rac/PAK and RAF/Janus kinase (JNK) signalling activation, strictly related to cell mobility [7];(2)activation of STAT3, a transcription factor with a fundamental role in normal cell growth, but also constitutively activated in about 70% of solid and haematological tumors [8];(3)PI3K/AKT activation pathway, which is normally hyperactivated in a lot of solid cancers. In fact, in recent years, in-depth studies have demonstrated that PI3K/AKT signalling is strictly related to cancer onset, tumor progression and proliferation, metastasis development, apoptosis, epithelial-mesenchymal transition, stem-like phenotype, immune microenvironment and drug resistance of cancer cells [9];(4)suppression of apoptosis p53-mediated and reduction of p53-transcriptional activity, pathways closely related to tumor survival, migration, invasion, and growth. In detail, through the ubiquitination process, FAK promotes p53 degradation, thus leading to tumor cell growth and proliferation [10].

## 2. FAK Role in Cancer Development

FAK hyper-activity causes resistance of cancer cells to traditional antitumor therapy [11]. The activation of downstream signalling enhances FAK expression in different solid malignancies, particularly ovarian cancer [12] and pancreatic cancer (PC) [13]. Additionally, FAK is up-regulated in a wide variety of solid cancers (such as head, oral, breast, neck, bladder, colorectal, lung, thyroid, prostatic, hepatocellular carcinomas) and also melanoma, osteosarcoma and glioblastoma [14]. For all these reasons, FAK can be considered an interesting target for cancer therapy, and development of FAK inhibitors (FAKIs) is one of the research hotspots for pharmaceutical companies, as well as academia.

## 3. FAK Inhibitors

A number of crystallographic structures of FAK kinase domain complexed with FAKIs has been published, thus, supporting the structure-based design of new FAKIs [15]. Particularly, small chemical molecules with good drug-like characteristics can inhibit FAK phosphorylation, thus blocking intracellular signalling through the cell membrane, and, finally, inhibiting proliferation of cancer cells. 

FAKIs could be classified into four groups: (1) inhibitors of ATP binding site (ATP competitive inhibitors); (2) inhibitors of FAK-FERM domain; (3) inhibitors of FAK-FAT domain; (4) allosteric inhibitors (non-ATP-competitive inhibitors). 

The majority of the most potent FAKIs so far identified target the ATP binding site in the kinase domain. In detail, the ATP-dependent FAKIs generally affect the ATP binding site of FAK, blocking FAK phosphorylation. These compounds generally bear a 2,4-substituted pyrimidine nucleus and, as assessed by X-ray crystallography, they share a common binding orientation with the nucleotide binding pocket of the kinase domain. In particular, the pyrimidine core interacts with the hinge region, by forming hydrogen bonds with the C=O of Cys502 and/or Glu500 residues, hydrophobic interaction with Ala452 and Leu553 and interaction with the helical DFG (Asp564, Phe565, Gly566) motif of the activation loop, whose conformation plays an essential role in obtaining FAK inhibition. In addition, in the kinase domain, gatekeeper residue (Met499) in the upper part and a solvent region in the lower part also have significant roles in the design of FAKIs.

Inhibitors of FAK-FERM domain prevent phosphorylation at Tyr397, whereas inhibitors of FAK-FAT domain hinder Tyr965 phosphorylation [16].

The ATP-competitive FAKIs exhibit good pre-clinical antiproliferative action against different solid tumors (such as glioma, ovarian, breast, oesophageal and gastrointestinal carcinomas). Moreover, some of them showed dose-dependent activity in different subcutaneous human xenograft models including breast, prostate, colon, and pancreatic cancers, and glioblastoma [17].

The non-ATP-competitive FAKIs bind an allosteric site and disrupt specific protein-protein interactions (e.g., p53-FAK interaction), causing more selective FAK inhibition. They evidenced some interesting in vitro and in vivo antiproliferative action, alone or in combination with other anticancer agents [18].

To date different FAKIs developed by different pharma companies (Pfizer, Novartis, GSK, Alchem Lab., Verastem Inc., Boehringer Ingelheim, Centaurus Biopharma Co., Ltd. and many others) are under evaluation in clinical and pre-clinical studies [16,17,18,19,20].

### 3.1. FAK Inhibitors in Clinical Development

FAKIs in clinical development belong to pyrimidine (PF-562271, Defactinib, Conteltinib, also named CT-707, CEP-37440, GSK-2256098, Table 1) or pyridine (VS-4718, Table 1) chemical classes. Only BI-853520 (also named IN-10018, Table 1) does not include a heterocycle core, but it is characterized by an acrylamido functionality. As previously reported, most of the FAKIs in clinical development are ATP-competitive inhibitors, such as Defactinib and PF-562271, whereas VS-4718 and GSK-2256098 are FAT-FERM inhibitors.

### 3.2. FAK Inhibitors in Pre-Clinical Development

As reported in Table 2, to date five compounds are in pre-clinical evaluation. In detail, TAE226, PF-573228 and PF-43196 are ATP-competitive inhibitors, containing the diaminopyrimidine scaffold, whereas Y15 (developed by Roswell Park Cancer Institute, USA) is a 1,2,4,5-benzenetetraamine tetrahydrochloride that directly inhibits FAK autophosphorylation at Tyr397 with IC_50_ of 1 μM [21]. In in vitro studies, Y15 enhanced cell detachment in thyroid cancer cell lines, and apoptosis in colon and breast cancer cells and melanoma. In vivo, pharmacodynamic and pharmacokinetic evaluations confirmed the significant antiproliferative activity of Y15, so that, therefore, it is considered an attractive FAKI candidate [22].

An in silico screening of the NCI database (more than 140,000 small molecules) allowed the identification of the tetraazaadamantane derivative Y11, which is able to inhibit FAK autophosphorylation on the FAK-FERM domain with an IC_50_ of 50 nM. This compound showed antiproliferative activity against breast BT474 and colon SW620 cancer cell lines [23]. 

Despite their chemical diversity, the compounds in pre-clinical development are ATP- competitive inhibitors and share a common binding mode within FAK active sites, as highlighted by the crystallographic analysis of FAK-TAE226 and FAK-PF573228 complexes (Figure 1).

### 3.3. FAK Inhibitors under Study

Starting from 2017 to date, a variety of chemical scaffolds have been investigated to obtain new FAKIs. Particularly, diphenylpyrimidines (DPPYs, the most studied), triazines, pyrrolo-pyrimidines, thieno-pyrimidines, oxadiazoles, triazoles and benzotriazoles, quinolines and thiosemicarbazone compounds have been tested for their activities against FAK (Table 3). Additionally, PROTAC molecules have been investigated.

#### 3.3.1. Diphenylpyrimidines (DPPYs)

The pyrimidine scaffold, especially 2,4-substituted, is, at the moment, the most investigated chemical scaffold by pharmaceutical companies and in academic research and has shown very interesting results. The most potent DPPY inhibitors share, with Defactinib (Table 1) and TAE226 (Table 2), the pyrimidine core and bear an N-methylbenzamide substructure, able to increase potency by mediating interactions with Cys502, Asp546, and Leu553 [24].

Liu and co-workers evidenced that the introduction of a sulphonamide function into the C-2 aniline of the pyrimidine core improved BTK inhibition of Spebrutinib [25]. Based on this consideration, a library of sulphonamide-substituted diphenylpyrimidine derivatives (Sul-DPPYs), structurally related to TAE226, were synthesized with the aim to increase activity against FAK. Some of these new Sul-DPPYs evidenced good activity against FAK (IC_50_ values lower than 100 nM) and compound **1** (Figure 2, IC_50_ = 86.7 nM) was able to block, in the low micromolar range, the proliferation of several classes of refractory cancer cell lines, including the PC cell lines (AsPC-1, Panc-1 and BxPC-3), NSCLC-resistant H1975 cell line, and the B lymphocyte cell line (Ramos cells). In addition, flow cytometry analysis indicated that **1** induced apoptosis of PC cells in a dose-dependent fashion, resulting in its being an interesting FAKI for the treatment of PC [26].

A library of soluble phosphamide-containing diphenylpyrimidine analogs (PA-DPPYs) were synthesized as potent FAKIs by Li and co-workers. These PA-DPPY derivatives inhibited FAK at concentrations lower than 10.69 nM in an enzymatic assay. Compounds **2a** and **2b** (Figure 2) were identified as the most active derivatives, with IC_50_ values of 4.25 nM and 4.65 nM, respectively. In addition, derivative **2b,** bearing a morpholinophosphoryl group, evidenced strong activity against the AsPC-1 cell line (IC_50_ = 1.66 μM), without cytotoxic effects. The authors also evidenced that the introduction of a phosphoryl group on aniline moiety strengthened FAK inhibition and anticancer activity against B-cell leukaemia cells. These interesting results pointed to compound **2b** being a valuable FAKI to treat PC [27]. 

Additional lead optimization studies confirmed that the introduction of H-bond acceptor groups (such as sulfonate and phosphate groups of compounds **1** and **2**) led to an increase of binding affinity for the FAK kinase domain. Thus, to obtain new FAKIs with strong hydrogen bonding affinity for the FAK enzyme, an additional *N*-morpholine amide function was inserted into the aniline moiety; in this manner, novel carbonyl-substituted diphenylpyrimidine derivatives (Car-DPPYs) were synthesized and evaluated. Compounds **3a** and **3b** (Figure 2) blocked FAK in enzymatic assay with comparable activity to TAE226 (IC_50_ = 5.17 nM and = 2.58 nM, respectively), showed potent anticancer activity against different cancer cells (AsPC-1 cells, BxPC-3 cells, and MCF-7/ADR cells) and displayed antiproliferative action in an in vivo xenograft mouse model. These interesting results confirmed Car-DPPYs as interesting pre-clinical candidates [28].

More recently, some 2,4-dianilinopyrimidine derivatives, containing 4-(morpholinomethyl)phenyl side chain, were reported as new FAKIs. Among the studied molecules, compound **4** (Figure 2) showed the best pharmacological profile, being able to block FAK at nanomolar level (IC_50_ = 47 nM). Additionally, this compound arrested the proliferation of different cancer cell lines (e.g., H1975 and A431 cells) at low micromolar concentration and caused apoptosis, arresting the cells in S/G2 phase. Furthermore, compound **4** blocked the migration of H1975 cells. From a chemical point of view, this derivative has a peculiar structure, being characterized by a benzamide portion bearing a hydroxyethyl chain as terminal tail. Overall, these results indicated **4** as a novel and promising FAKI that can be further studied as an anticancer agent [29].

Other DPPYs, synthesized by Wang and co-workers, suppressed FAK activity and angiogenesis, **5a,b** (Figure 2) being the most potent compounds (IC_50_ values of 2.75 and 1.87 nM, respectively). Derivatives **5** showed strong anticancer effects against different human cancer cells, including two FAK-overexpressing PC cells (PANC-1 and BxPC-3). In addition, they suppressed colony formation, migration and invasion of PANC-1 cells, also inducing apoptosis and G2/M phase cell cycle arrest. Western blot analysis confirmed that these compounds blocked the FAK/PI3K/AKT signalling pathway and effectively decreased cyclin D1 and Bcl-2 expression. In addition, **5** potently altered human vascular endothelial cell (HUVEC) morphology and inhibited the migration and tube formation of these cells. Consistently, these compounds exerted antiangiogenetic activity in a zebrafish model, thus, supporting the potential of this class of molecules as antiangiogenic agents [30].

Ai and co-workers reported a new class of substituted DPPYs as potential dual FAK/EGFR^T790M^ inhibitors. EGFR (Endothelial growth factor receptor) is a well-known target for anticancer therapies, especially for non-small lung cancer (NSCLC) treatment, but the T790M mutation in the EGFR ATP site has recently caused a lack of EGFR-targeted therapy. To circumvent this problem, the pyrimidine scaffold of TAE226 was modified by a fragment-based drug design approach and different chemical entities typical of EGFRT^790M^ inhibitors were inserted. In the so obtained dual FAK/EGFRT^790M^ inhibitors the morpholino substituent is not directly linked to aniline moiety as in TAE226, but separated by an amide linker. Most of the prepared compounds evidenced interesting antiproliferative activity against FAK-overexpressing PC cells (namely, AsPC-1, BxPC-3, Panc-1) and also against two drug-resistant cancer cell lines (breast cancer MCF-7 cells and lung cancer H1975 cells) at concentrations lower than 7 μM. Compounds **6a** and **6b** (Figure 2) potently inhibited FAK with IC_50_ values of 1.03 and 3.05 nM, respectively, and also EGFRT^790M^ (IC_50_ 3.89 and 7.13 nM, respectively); in addition, **6a** was effective in an in vivo AsPC-1 cell xenograft mouse model. In conclusion, this study represents a new outlook for the treatment of hard-to-treat cancers [31].

Following a molecular hybridization approach, DPPYs containing dithiocarbamate moiety were designed and synthesized. Particularly, derivative **7** (Figure 2) resulted in a very potent FAKI (IC_50_ = 0.07 nM), blocking HCT116, PC-3, U87-MG and MCF-7 cell proliferation (IC_50_ values ranging from 0.001 μM to 0.06 μM), without cytotoxic effect, in a non-malignant cell line (MCF-10A). Moreover, compound **7** induced cell cycle arrest at the G2/M phase and apoptosis in both HCT116 and MCF-7 cells. Other biological studies showed that **7** inhibited MCF-7 cell migration and exerted an anti-angiogenesis effect on HUVECs [32].

To evaluate their potentiality as positron emission tomography (PET) imaging agents in cancer detection, novel pyrimidine derivatives were designed by computer-assisted drug design and successfully synthesized and characterized. These new derivatives were tested as FAKIs, resulting in **8a** (Figure 2) being the most interesting (IC_50_ = 0.06 μM). In addition, four of these compounds were successively labelled with positron emitter ^18^F, showing proper log *p* values and high stability in mouse plasma. In particular, [^18^F]-labelled compound **8b** (Figure 2), characterized by a 4-OCH_3_ on the benzene ring, was well evidenced in in vivo biodistribution in bearing S180 tumor mice, suggesting that it might be a new valuable probe for PET cancer imaging [33]. More recently, a series of novel [^18^F]-labelled 2,4-diaminopyrimidine were designed and synthesized as FAKIs. The whole series showed good FAK inhibition (IC_50_ ranging from 5.0 to 205.1 nM) and compounds **9a** and **9b** (Figure 2) exhibited IC_50_ values of 5.0 nM and 21.6 nM, respectively, and good biodistributions. In particular, **9a** revealed promising target-to-non-target ratios, with tumor/blood, tumor/muscle, and tumor/bone ratios of 1.17, 2.99 and 2.19, respectively, 30 min after injection. These results indicated that **9a** could be a potential PET tracer for cancer diagnosis [34]. More recently, the same authors reported other ^18^F-labelled 2,4-diaminopyrimidine derivatives as PET tumor imaging agents. The new compounds were close analogs of **9**, TAE226 and PF-562271 and inhibited FAK enzyme with IC_50_ values in the nanomolar range. Compound **10** (Figure 2) was identified as the most potent derivative (IC_50_ = 3.2 nM) and evidenced good biodistribution in S180-bearing mice, so supporting its value as a PET imaging agent [35].

Xie and Chen reported pyrimidine derivatives in which the C-2 anilino and C-5 positions were replaced by different heterocycles. In particular, the insertion of a differently decorated pyrazole nucleus on C-2 position led to potent FAKIs (IC_50_ < 1 nM, comparable to GSK-2256098) that proved to be active against different cancer cell lines (MDA-MB-231, BXPC- 3, NCI-H1975, DU145 and 786O). Among these derivatives, compound **11** (Figure 2) showed the best enzymatic profile, being able to inhibit FAK Y397 phosphorylation in MDA-MB-231 cell line and to induce apoptosis. In addition, computational docking studies evidenced that **11** and TAE226 share a similar binding orientation within the catalytic pocket of the FAK kinase domain [36]. More recently, Chen and co-workers designed and synthesized a small library (28 compounds) of FAKIs characterized by a phenoxy substituent on C-5 position of the pyrimidine moiety. The most promising compound (**12**, Figure 2) displayed good potency (IC_50_ = 45 nM) and selectivity against FAK, and evidenced potent antiproliferative activity against Hela, HCT116 and MDA-MB-231 cell lines. In addition, **12** counteracted clone formation, HCT-116 cell migration and HUVEC tube formation; furthermore, this compound caused cell cycle arrest in the G2/M phase, causing apoptosis by promoting reactive oxygen species (ROS) production and blocking the FAK-Src-ERK (Extracellular signal-regulated kinases) signalling pathway in a dose-dependent manner. Moreover, **12** was endowed with good oral bioavailability and inhibited tumor growth in the HCT116 xenograft model [37].

Irreversible sub-nanomolar FAKIs (derivatives **13**–**16**, Figure 3) are characterized by a pyrimidine scaffold and a Michael acceptor able to act as an electrophile and covalently modify the enzyme. Furthermore, the majority of these derivatives bear an additional 4-membered cycle inserted in the para position of a phenyl ring in the C-2-position of the pyrimidine core. The FAK-**13** crystallographic complex (PDB ID: 6GCX) is mainly stabilized by the covalent bond with Cys427 (Figure 4). Further stabilizing interactions include three hydrogen bonds occurring between the ligand and Cys502 backbone in the kinase hinge region and Asp564 backbone of the DFG motif (Figure 4). In addition, some hydrophobic interactions are evidenced between the C atoms of the 2-aniline ring and Ile428 and Gly505, and between C atoms of the pyrimidine ring and Ala452 and Leu553 side chains. Compound **13** potently inhibited FAK autophosphorylation in SCC cells, suggesting that it could significantly block the intracellular FAK signalling pathway. In particular, in SCC cells, **13** evidenced an antiproliferative action similar to previously reported FAT-FERM inhibitor VS-4718. Moreover, this compound evidenced low inhibition against the insulin receptor (IR) kinase, providing a new model to overcome the side effects reported by TAE226 [38].

More recently, the same authors reported a library of DPPYs as useful agents against human malignant glioblastoma (GBM), an invasive brain tumor in which FAK is over-expressed and hyper-activated. Compounds **14** (Figure 3) were identified as novel irreversible FAKIs with good potency against FAK (IC_50_ values ranging from 0.6 nM to 16.3 nM) and able to retard tumor cell growth in GBM cell lines (U-87 MG, U251, and A172 cell lines), strongly reducing U-87 cell migration and cell cycle progression. Furthermore, in GBM cells, compounds **14** induced a relevant decrease of FAK autophosphorylation and of its downstream signalling (e.g., AKT, ERK and nuclear factor-κB, NF-kB). All these data evidenced the potential therapeutic benefits of covalent FAKIs, that could represent an interesting new targeted therapy for the GBM treatment [39]. 

More recently, Zhang and co-workers reported derivative **15** (2-((2-((4-((2-((2-acrylamidoethyl)amino)-3,4-dioxocyclobut-1-en-1-yl)amino)phenyl)amino)-5-(trifluoromethyl)pyrimidin-4-yl)amino)-*N*-methylbenzamide, Figure 3) as a new irreversible DPPY FAKI. This compound is closely related to derivative **13**; the only difference between the two compounds is the methylene group connecting the aniline core with the dioxocyclobutenyl moiety. In enzymatic assay, derivative **15** inhibited FAK activity at nanomolar concentration (IC_50_ value of 5.9 nM) and in vitro showed strong antiproliferative activity against four human cancer cell lines (U-87 MG, MDA-MB-231, PC-3, and MCF-7), in which FAK is over-expressed [40].

TAE226 was selected as the lead structure for the preparation of a large library of irreversible covalent FAKIs. Compound **16** (Figure 3) was identified as the most interesting molecule of the series, being able to inhibit FAK at nanomolar concentration (IC_50_ = 35 nM) and showing potent antiproliferative activity against Hela (IC_50_ = 410 nM), HCT116 (IC_50_ = 10 nM) and MDA-MB-231 (IC_50_ = 110 nM) cell lines. Furthermore, **16** inhibited HCT-116 clone formation and migration and blocked cell cycle arrest in the G2/M phase. Docking simulations speculated the ability of the compound to covalently bind FAK Cys427 residue. Interestingly, **16** evidenced good oral bioavailability and growth inhibition in the HCT116 tumor xenograft model. For all these reasons, **16** could represent a promising covalent FAKI [41].

A recent Chinese study reported the preparation of eight 2, 4-disubstituted-5-(trifluoromethyl) pyrimidine compounds (formula not disclosed) able to block FAK at nanomolar levels (IC_50_ = 6 nM) and to inhibit proliferation of different cancer cell lines (U87-MG and A549) at micromolar concentration [42].

#### 3.3.2. 1,2,4-Triazines

In the recent past, 1,3,5-triazine derivatives have been extensively studied as FAKIs. X-ray crystallographic analyses confirmed that these compounds share with TAE226 a similar interaction mode with the FAK kinase domain [43,44]. To further explore this heterocyclic scaffold, the triazine core was fused with imidazo one, to obtain imidazo-triazine derivatives with nanomolar potencies against FAK [45]. 

More recently, 1,2,4-diarylaminotriazines **17** (Figure 5) were identified as micromolar FAKIs, able to block glioblastoma (U-87MG) and colon (HCT-116) cancer cell line proliferation with good potency. Compounds **17** showed lower in vitro potency than TAE226. According to docking simulations, the insertion of an extra nitrogen atom in the triazine scaffold would result in less electrons being available for hydrogen bonding with the FAK hinge region, therefore reducing the stability of the complex [46]. 

#### 3.3.3. 7*H*-Pyrrolo [2,3-*d*]pyrimidines

To discover novel chemical classes of FAKIs, He and colleagues introduced a five-membered ring to bridge positions 4 and 5 of the pyrimidine [47]. The so obtained bicyclic pyrrolo[2,3-*d*]pyrimidines were further modified and the introduction of a dimethylphosphine oxide moiety led to compound **18** (Figure 6), a nanomolar FAKI (IC_50_ = 5.4 nM) that showed interesting selectivity profile, relevant anticancer activity against A549 cells (IC_50_ = 3.2 μM) and low cytotoxicity. Additional studies evidenced good in vitro metabolic stability in mouse, rat and human liver microsomes for **18**, and reduced inhibitory activity against human cytochrome P450 [48].

Subsequently, the same authors identified compound **19**, (Figure 6) a multi-target kinase inhibitor, which is a close analog of TAE226. Compound **19** potently inhibited FAK (IC_50_ = 19.1 nM) and induced apoptosis of MDA-MB-231 cells. Furthermore, it blocked the proliferation of different cancer cell lines (U-87MG, A-549 and MDA-MB-231 cells), resulting in its being more active than TAE226. Interestingly, **19** exhibited low cytotoxicity toward normal human cell line HK2. FAK-**19** docking complex evidenced the following: (1) the pyrrolo nucleus penetrates into the ATP binding pocket, making van der Waals interactions with the gatekeeper residue Met499; (2) three hydrogen bonds are present between the N atoms of the pyrrolo-pyrimidine core and Glu500 and Gly502 of the kinase hinge region; (3) the benzamide substituent forms a hydrogen bond with Asp564 of the DFG motif; (4) the *N*-methylpiperazine tail points toward the solvent region; (5) additional hydrophobic interactions are present between C atoms of pyrrolo-pyrimidine core and Leu553 and Ala452, and also between the C of the 2-aniline ring and Gly505 and Glu506. All these data evidenced that **19** could represent a novel lead compound for the development of FAK-targeted anticancer therapy [49].

More recently, on the basis of docking studies performed on a library of 667 fragments into the ATP pocket of the FAK-PF-562271 crystallographic complex (PDB ID: 3BZ3), novel 7*H*-pyrrolo[2,3-*d*]pyrimidines were reported. These derivatives, bearing an additional fluorine atom on position 5 of the central scaffold, evidenced good activity against FAK and blocked the proliferation of different cancer cell lines (SMMC7721 and YY8103). Particularly, compound **20** (2-((2-((3-(acetamidomethyl)phenyl)amino)-5-fluoro-7*H*-pyrrolo[2,3-*d*]pyrimidin-4-yl)amino)-N-methylbenzamide, Figure 6) was selected for additional investigations, including preliminary pharmacokinetic studies in rats, in vivo toxicity assays in mice and a xenograft tumor model. Results indicated that **20**, at 200 mg/kg dose, did not change rat body weight, and strongly decreased tumor growth. Furthermore, additional analyses suggested that **20** inhibited hepatocellular carcinoma cell (HCC) proliferation, blocking phosphorylation in FAK intracellular signalling [50].

As reported before, the FAK role in ovarian cancer development has been deeply demonstrated. In fact, FAK over-expression and high p-FAK have been evidenced in a majority of ovarian cancer cases [51]. In addition, in ovarian cancer, FAK over-expression has been correlated with poor survival, high-stage tumors and metastasis formation [52]. To identify novel agents for ovarian cancer treatment, Wei and co-workers, using a scaffold hopping strategy, discovered new FAKIs. New 7*H*-pyrrolo[2,3-*d*]pyrimidine **21** (Figure 6) showed potent activity (IC_50_ = 1.89 nM) and good selectivity for FAK and was endowed in vitro with good solubility and metabolic stability. Derivative **21** reduced PA-1 cell migration and invasion, decreased MMP-2 and MMP-9 expression and, when orally administered to mice, suppressed ovarian cancer growth and metastasis formation, without adverse effects. All these biological results underlined the potential of this compound for ovarian cancer treatment [53].

#### 3.3.4. Thieno[3,2-*d*]pyrimidines

The isosteric N/S substitution on the pyrrolo[2,3-*d*]pyrimidine nucleus led to the identification of 2,7-disubstituted-thieno[3,2-*d*]pyrimidines as novel FAKIs (compounds **22** and **23**, Figure 6). Compound **22** suppressed FAK activity (IC_50_ = 28.2 nM) and inhibited U-87MG (IC_50_ = 0.16 μM), A-549 (IC_50_ = 0.27 μM) and MDA-MB-231(IC_50_ = 0.19 μM) cancer cell line proliferation. In addition, **22** exhibited relatively less cytotoxicity (IC_50_ = 3.32 μM) toward a normal human cell line (HK2), reduced MDA-MB-231 cell migration, induced apoptosis and caused cellular arrest in G0/G1 phase. As reported in Figure 6, docking calculations evidenced that **22** is anchored to the hinge region by a double-dentate hydrogen bond between the aminopyrimidine moiety and Cys502 backbone; additionally, the OCH_3_ substituent establishes a hydrogen bond with Asp564 of the DFG motif, whereas the piperidine tail interacts with Cys427, confirming its important role in enzymatic inhibition. This evidence confirmed that **22** could represent a promising lead as a FAKI [54].

Through an intensive SAR study, derivative **23** (Figure 6) was identified as a novel multitarget rigid analog of PF-562271 (Table 1). Compound **23** showed potent activity against 15 kinases, including FLT3 and FAK (IC_50_ = 9.7 nM) and possessed excellent potency against both FLT3 (including F691L) and FAK mutants. Moreover, this derivative proved to be more effective than PF-562271 in apoptosis induction, anchorage-independent growth inhibition, and tumor reduction in the MDA-MB-231 xenograft mouse model. In addition, **23** caused regression of tumor growth in the MV4−11 xenograft mouse model, evidencing its possible action in acute myeloid leukemia (AML). Finally, **23** significantly prevented metastasis of orthotopic MDA-MB-231 cell tumor. All these biological data support the therapeutic potential of this thieno[3,2-*d*]pyrimidine for the treatment of highly invasive cancers, including drug-resistant AML harboring recalcitrant FLT3 and/or FAK mutants [55].

#### 3.3.5. 1,3,4-Oxadiazoles

1,4-Benzodioxan scaffold emerged as an important pharmacophore with various pharmaceutical applications, including anticancer and anti-inflammatory therapy [56]. Consequently, some authors have fused this scaffold with 1,3,4-oxadiazole one, chosen because it has favourable properties in respect to other heterocycle nuclei. In 2017, a small library (17 compounds) of 1,3,4-oxadiazole-2(3*H*)-thione derivatives, bearing 1,4-benzodioxan moiety and an additional piperazine tail, were synthesized and tested on four cancer cell lines (HepG2, Hela, SW116 and BGC823). In particular, compound **24** (Figure 7) was identified as the most promising derivative of the series with a FAK IC_50_ values of 0.78 μM. According to molecular docking calculations, compound **24** could bind the FAK-ATP binding site; the FAK-**24** docking complex would be mainly stabilized by a hydrogen bond between the 1,4-benzodioxan oxygen and Asp564 backbone and an electrostatic interaction involving the piperazine nitrogen (in the protonated form) and Glu506 side chain carboxylate. Overall, the computational results suggested that 1,4-benzodioxan and piperazine moieties could represent relevant substructures for the development of novel and potent FAKIs [57].

Other 1,3,4-oxadiazole derivatives **25** (Figure 7) were synthesized, tested on A549, C6 and NIH/3T3 cell lines and, subsequently, investigated for their effects on apoptosis, caspase-3 activation, AKT, FAK and MMP. Out of the evaluated derivatives, three compounds showed optimal pharmacological profiles, being more active than cisplatin and significantly inhibiting FAK phosphorylation activity in the C6 cell line (no enzymatic assay has been reported). Docking simulations evidenced that the most active derivatives could fit into the kinase domain of both AKT and FAK with high affinity. These findings further support the potential of 1,3,4-oxadiazole scaffold for the discovery of new FAKIs [58].

#### 3.3.6. 1,2,4-Triazole and Benzotriazole Derivatives

In 2021, Mustafa and co-workers designed and synthesized twenty new dual HDAC2/FAKIs, merging the interesting biological properties of 1,2,4-triazole with those of the pyridine ring. These 1,2,4-triazoles bear a 5-pyridinyl side chain and showed good activity as histone deacetylase (HDAC) inhibitors and FAKIs. Compound **26** (Figure 7) was the most active, with IC_50_ values of 90 nM and 12.59 nM for HDAC2 and FAK, respectively. This triazole derivative showed FAK inhibitory properties similar to those of TAE226 and was endowed with a good selectivity profile. In addition, **26** showed antiproliferative action on A-498 and Caki-1 renal cancer cells, being more active than vorinostat and valproic acid, used as reference compounds. In the same cell lines, **26** strongly arrested the cell cycle at the G2/M phase and triggered apoptosis. Taken together, these results highlight the potential of this novel triazole derivative as an interesting dual HDAC2/FAK inhibitor [59]. Further SAR extension studies led to the identification of four 1,2,4-triazole derivatives, in which the methyl group in N4 position was replaced by a phenyl ring and the hydroxamic acid in the side chain was modified. The new derivatives were able to block FAK and were endowed with antiproliferative activity against liver cancer cells (HepG2 and Hep3B). In enzymatic assays, compound **27** (Figure 7) proved to be more potent than reference compound GSK-2256098 in inhibiting FAK. Furthermore, **27** blocked FAK phosphorylation in HepG2 cells and decreased PI3K, AKT, JNK, and STAT3 phosphorylation levels, leading to cell cycle arrest and apoptosis induction. Overall, these biological data indicate that **27** could represent a potential candidate for cancer treatment [60].

The adoption of a fragment-based design strategy allowed the identification of a new series of benzotriazole-containing acylarylhydrazones as novel FAKIs. In these molecules, the benzotriazole scaffold acts as H bond acceptor in the FAK kinase domain, whereas the N-acylhydrazone scaffold could chelate zinc ion necessary for caspase activation. All compounds were tested for their antiproliferative activity against 60 human tumor cell lines (NCI, Bethesda, USA), showing activity in some cases higher than doxorubicin, used as reference compound. The most active derivatives inhibited FAK at nanomolar concentrations and **28** (Figure 7) was the most promising compound, being able to block FAK and proline-rich tyrosine kinase 2 (Pyk2) activity with IC_50_ values of 44.6 and 70.19 nM, respectively. Additionally, it displayed inhibition of FAK phosphorylation in cell-based assay and caused significant increase in caspase-3 activity, cell accumulation in pre-G1 phase and cell cycle arrest at G2/M phase, indicating an apoptotic mechanism [61].

#### 3.3.7. Arylquinolines

Very recently, Elbadawi reported 2-arylquinolines and 2,6-diarylquinoline derivatives as new dual EGFR/FAKIs to fight cancer by a non-overlapping downstream signalling/inhibition [62], as previously demonstrated by Ai [31]. The modification of arylquinoline Topoisomerase I (TOP1) inhibitors, endowed with good antitumor activity, allowed the synthesis of a new library of quinoline derivatives, structurally related to gefitinib, with potent antiproliferative effect against colorectal cancer cell lines (DLD-1 and HCT-116), but characterized by weak TOP1 inhibition. Screening against a panel of nine kinases evidenced their EGFR/FAK dual inhibitory activity, with compound **29** (Figure 7) being the most interesting analog (IC_50_ values for FAK and EGFR of 14.25 nM and 20.15 nM respectively). Molecular docking and molecular dynamics simulation confirmed that this derivative represented the first quinoline reported as being a dual EGFR/FAK inhibitor [63].

#### 3.3.8. Thiosemicarbazones

A small library of six novel bis(thiosemicarbazone)copper(I) complexes showed good antiproliferative action on different tumor cell lines (i.e., human breast adenocarcinoma MCF-7, cervical HeLa, epithelioma Hep-2 and Ehrlich ascites carcinoma EAC). Despite no specific enzymatic assay on the FAK enzyme being reported, the authors carried out in silico docking studies on derivative **30** (Figure 7) to define its binding mode within the FAK-ATP binding site [64].

## 4. PROTAC Molecules

PROTAC (Proteolysis targeting chimera) molecules represent an innovative medicinal chemistry approach based on the development of bifunctional entities that cause post-translational knockdown of a target protein by E3 ubiquitin ligase and subsequent proteasomal degradation. To date, only four examples of PROTACs for FAK have been reported.

In detail, Cromm and co-workers changed the structure of Defactinib, replacing the N-methyl benzamide substructure with a 4-aminophenoxy group, to facilitate linker attachment and to increase cellular permeability. The pyrazine core of Defactinib was replaced by a 1,3-substituted aromatic ring, previously reported as having an interesting FAK inhibitor moiety. Through an amide linker, this new chemical entity has, therefore, linked to von Hippel-Lindau (VHL) E3 ligase, obtaining a series of PROTAC molecules. The most active compound (Figure 8) showed an IC_50_ value of 6.5 nM in the FAK inhibition test and a DC_50_ value of 3 nM in FAK degradation (Western blot analysis on PC3 cells), showing an increased selectivity in comparison with Defactinib [65].

Other researchers at Boehringer Ingelheim RCV GmbH & Co reported two highly selective and functional FAK-PROTACs based on von Hippel−Lindau and cereblon ligands, using BI-3663 and BI-0319 scaffolds (Figure 8). These PROTACs have been tested on a panel of HCC (11 cell lines), tongue squamous cell carcinoma, melanoma, pancreatic ductal adenocarcinoma, and NSCL cancer cell lines, in which FAK is over-expressed. Particularly, both PROTACs were analyzed for in vitro FAK engagement, ligase dependence selectivity and degradation efficacy in twelve cell lines (one lung cancer and 11 HCC cell lines). These cereblon-based compounds degraded FAK with a DC_50_ of about 30 nM on a panel of eleven HCC cell lines, but, unfortunately, despite their good action on FAK depletion, both were not able to block proliferation of the selected cancer cell lines. These results suggested that FAK depletion could be insufficient to affect cell proliferation; nevertheless, both reported PROTACs provide valuable tools to effectively reduce FAK levels in experimental conditions that might be better suited to reveal and differentiate between kinase-dependent and -independent FAK functions. In fact, although different studies have demonstrated that FAK can be targeted by a PROTAC approach, it has been clearly highlighted that there is a need for more selective FAK-PROTACs to address their role in cancer [66].

More recently, Gao and colleagues reported new PROTAC molecules, in which PF-562271 and Defactinib were linked to CRBN (Pomalidomide) E3 ligand. In detail, the lactam ring of PF-562271 does not play a fundamental role in protein binding interaction, therefore representing a suitable site to link with ligands of E3 ligase. Based on these considerations, the authors combined PF-562271 and Defactinib, various linkers and the ligand of CRBN-based E3 ligase. In detail, one PROTAC derivative (Figure 8) showed a quick and reversible FAK degradation with a DC_50_ in a picomolar range on various cancer cell lines [67].

More recently, research by GlaxoSmithKline reported the design and characterization of a new PROTAC molecule (Figure 8), based on a binder for the VHL E3 ligase and GSK215, a FAKI strictly related to previous reported VS-4718. This new PROTAC entity resulted in a potent and selective in vitro and in vivo FAK degrader molecule, with a different biological behavior in respect to classical FAKIs in cancer cells. The high FAK degradation seems to be related to a highly cooperative ternary complex, as evidenced by X-ray crystallography studies (Figure 9). In detail, this PROTAC molecule in mouse liver was able to degrade FAK at low doses and in a rapid and prolonged manner. This study should be useful for the study of FAK-degradation biology in vivo, confirming that FAK-PROTACs could be potential therapeutic agents to treat FAK-related diseases [68].

## 5. Conclusions

In recent years, FAK has represented a new interesting target in medicinal chemistry research and, consequently, a lot of FAKIs have been patented [69,70,71,72] and reported in the literature. In the last five years, many promising chemical scaffolds have been developed, in association with in silico strategies, to obtain new potent FAKIs.

Unfortunately, no compounds were launched on the market. The most promising derivatives, now in phase II, are Defactinib, Conteltinib and GSK-2256098. Other derivatives (e.g., TAE226) are under pre-clinical investigations and largely studied in different solid tumors. All compounds under current investigation are ATP-competitive inhibitors, but high potency and good selectivity could also be obtained using type II inhibitors (that target DFG-out conformation) or allosteric inhibitors. In addition, PROTAC molecules represent a promising approach for anticancer therapy.

Recently, many efforts have been made to investigate compounds with activity not only on the FAK enzyme, but also on other intracellular kinases involved in cancer progression, such as Pyk2 (as PF-373228), IGF-1R Insulin-like growth factor-I receptor (IGF-1R as TAE-226), EGFR [31], Anaplastic lymphoma kinase (ALK as CEP-37440) [16] and, very recently, molecules able to simultaneously target Btk and FAK have appeared in the literature [73].

Furthermore, targeting FAK and other proteins involved in its intracellular pathways (such as Vascular endothelial growth factor receptor, VEGFR, signal) has emerged as a new and interesting multi-target strategy to be kept under observation in the coming years. In fact, anticancer small molecules, able to specifically interact with a single intracellular target (such as FAKIs), have shown therapeutic disadvantages (e.g., rapid resistance onset) that limit their clinical efficacy. On the contrary, small molecules endowed with the capacity to simultaneously target two or more proteins involved at intracellular level in the same signalling pathways, such as VEGFR/FAK/Src/PI3K, might represent an alternative approach to treat cancer, in association with recombinant-protein and selective kinase inhibitors. In this context, the authors reported some pyrazole compounds endowed with anti-angiogenetic activity in HUVEC cells [74,75], which act on calcium mobilization, F-actin localization and FAK expression [76,77]. The importance of FAK for integrin functions suggests that these pyrazole compounds can be efficient inhibitors of cell adhesion, migration and extracellular matrix synthesis, thus, potentially broadening their target list to cancer-associated fibroblasts, tumor invasion and metastasis.

For all these reasons, the search of novel FAKIs is still of great interest for academia and pharmaceutical companies, representing a new emerging therapeutic option for cancer treatment.

## Figures and Tables

**Figure 1 ijms-23-06381-f001:**
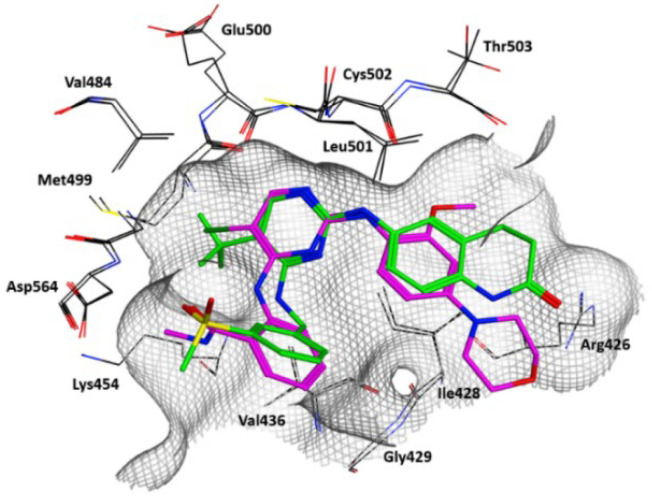
Superposition of the FAK-TAE226 (PDB code: 2JKK) and FAK-PF573228 (PDB code: 6YOJ) complexes. The carbon atoms of the two ligands are colored green (PF573228) and purple (TAE226). Carbon atoms of the protein are displayed in black. Oxygen, nitrogen, chlorine and sulfur atoms are represented in red, blue, dark green and yellow, respectively. Relevant residues lining the binding site are displayed. The molecular surface of the binding is shown as a grey grid.

**Figure 2 ijms-23-06381-f002:**
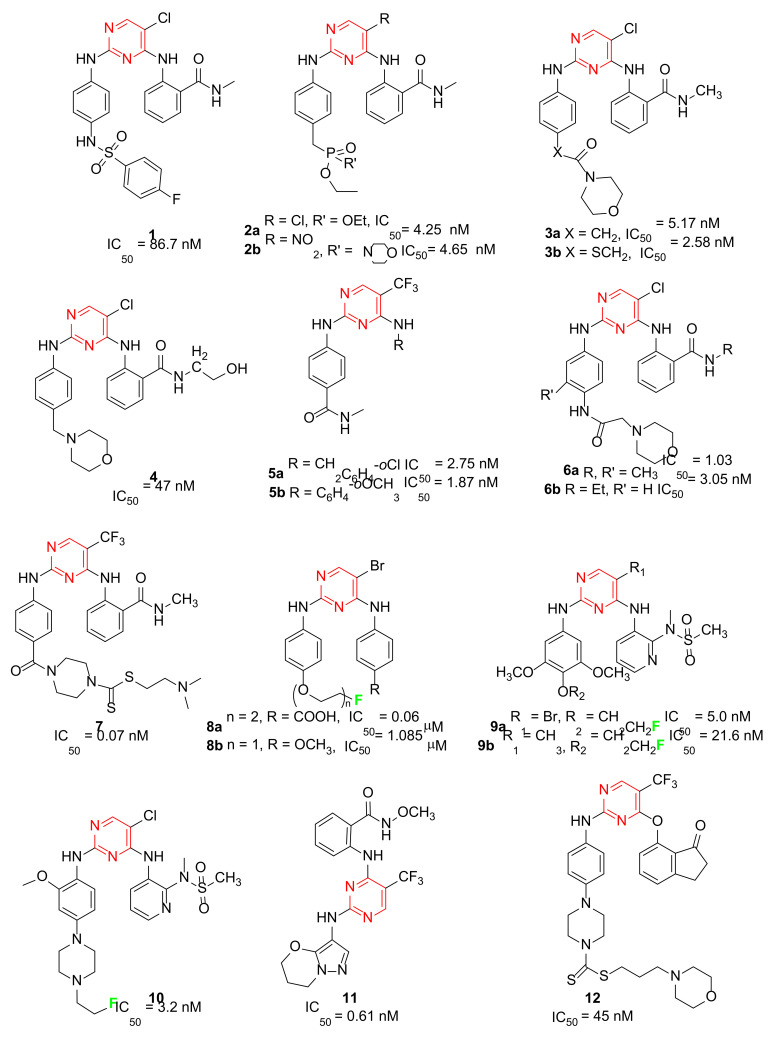
2,4-Diphenylpyrimidines (DPPY) as FAK inhibitors. The pyrimidine nucleus and labelled fluorine atom are reported in red and green, respectively.

**Figure 3 ijms-23-06381-f003:**
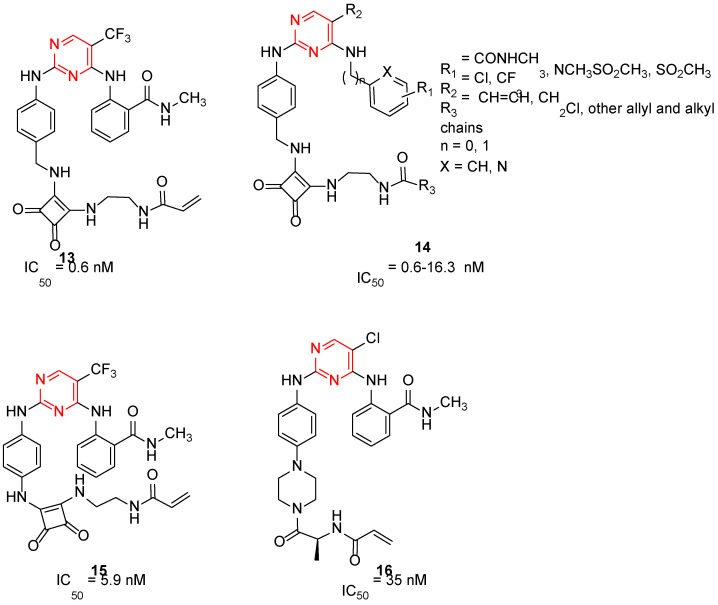
Irreversible pyrimidines as FAK inhibitors. The pyrimidine nucleus is reported in red.

**Figure 4 ijms-23-06381-f004:**
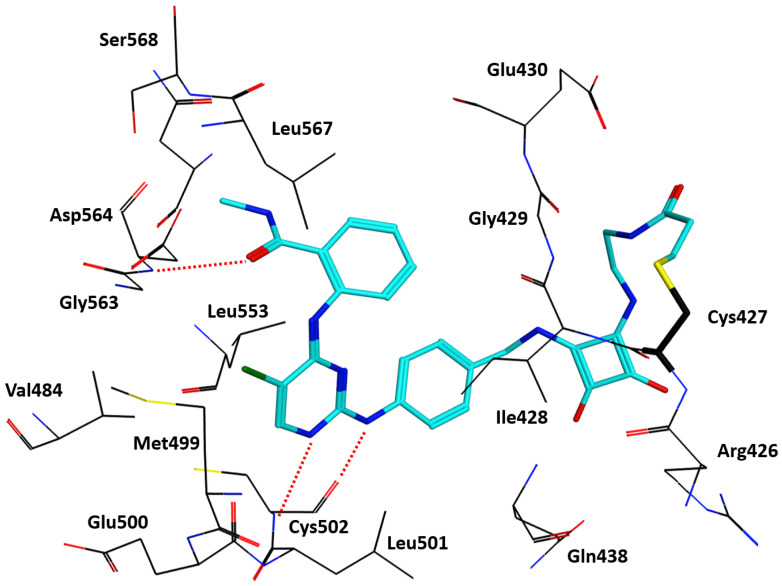
FAK-**13** crystallographic complex (PDB code: 6GCX). The ligand carbon atoms are coloured cyan. Carbon atoms of the protein are displayed in black. Oxygen, nitrogen and sulfur atoms are represented in red, blue and yellow, respectively. The three hydrogen bonds stabilising the complex are represented as dotted red lines. The covalently modified Cys427 is represented in a stick model. Selected residues limiting the binding site are shown.

**Figure 5 ijms-23-06381-f005:**
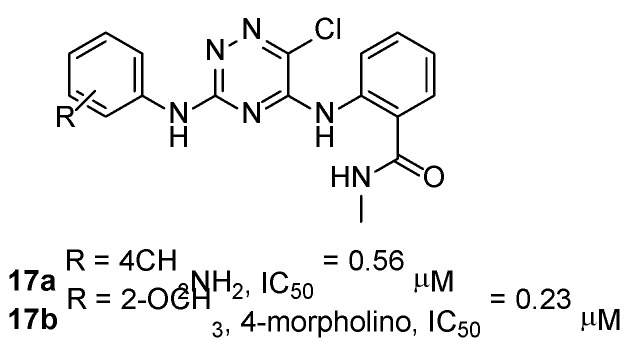
Chemical structure of triazine compounds as FAK inhibitors.

**Figure 6 ijms-23-06381-f006:**
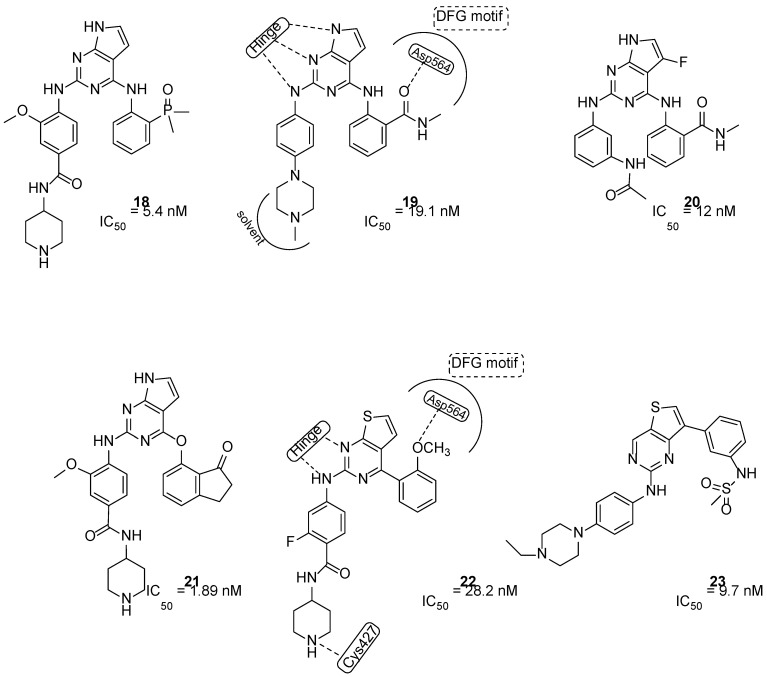
Structures of 7*H*-pyrrolo[2,3-*d*]pyrimidines and thieno[3,2-*d*]pyrimidines as FAK inhibitors.

**Figure 7 ijms-23-06381-f007:**
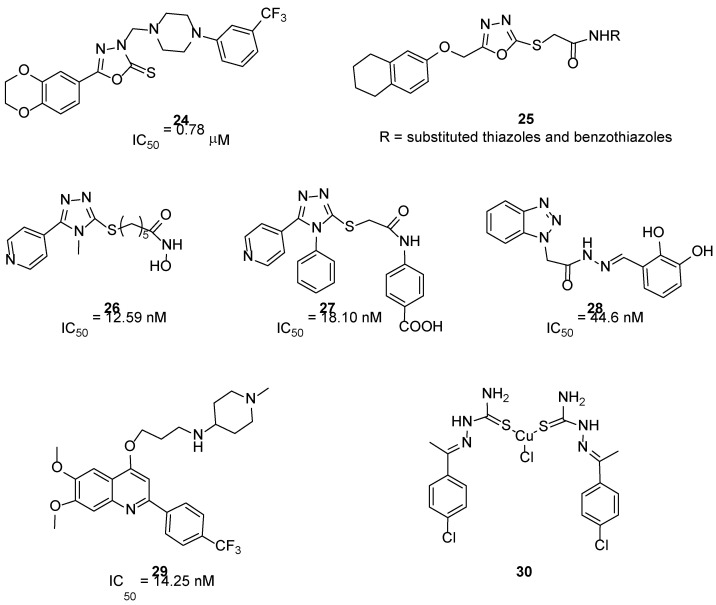
Other miscellaneous compounds as FAK inhibitors.

**Figure 8 ijms-23-06381-f008:**
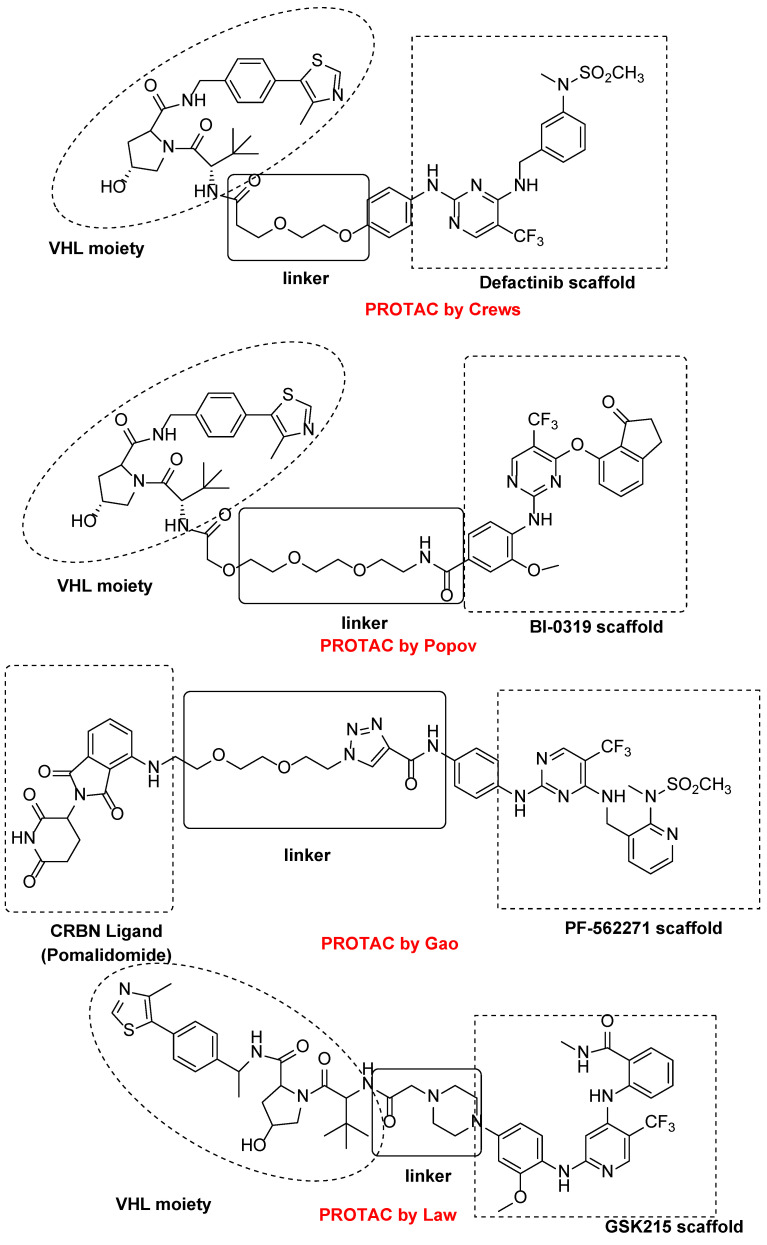
PROTAC molecules as FAK inhibitors.

**Figure 9 ijms-23-06381-f009:**
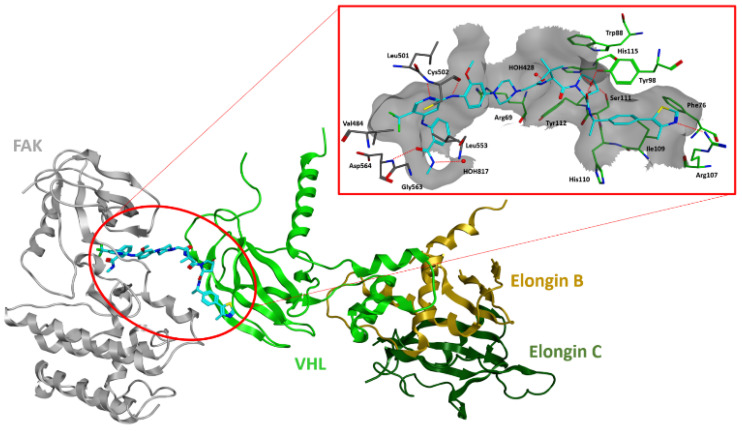
PROTAC- GSK215 in complex with FAK and pVHL:ElonginC:ElonginB (PDB code: 7PI4). The macromolecules forming the complex are differently colored as indicated. The carbon atoms of PROTAC ligand are reported in cyan. A zoom view of the ligand binding site is reported in the panel. FAK residues are colored grey, whereas VHL residues are shown in green. Oxygen, nitrogen, fluorine and sulfur atoms are represented in red, blue, light green and yellow, respectively. The hydrogen bonds between the ligand and the proteins are indicated as dotted red lines. Relevant water molecules are reported as red dots.

**Table 1 ijms-23-06381-t001:** Name, chemical structure, clinical trials and IC_50_s of compounds in clinical studies. The pyrimidine and pyridine scaffolds are evidenced in red and in green, respectively.

Name	Chemical Structure	Phase	Number of Clinical Trial	IC_50_ on FAK
Defactinib	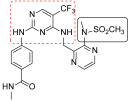	II	21	0.6 nM
PF-562271 VS-6062	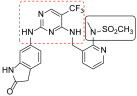	I	1	1.5 nM
CEP-37440	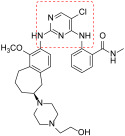	I	1	2 nM
Conteltinib (CT-707)	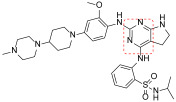	II	2	1.6 nM
VS-4718	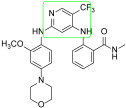	I	3	1.5 nM
GSK-2256098	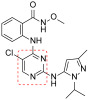	II	6	1.5 nM
BI-853520	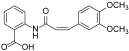	I	3	1 nM

**Table 2 ijms-23-06381-t002:** Name, chemical structure, clinical trials and IC_50_s of compounds in pre-clinical studies.

Name	Chemical Structure	Type of Inhibitors	Cancer Type	IC_50_ on FAK
TAE226	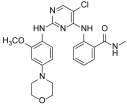	ATP competitive inhibitor	Pancreatic, prostatic, head, neck cancer	5.5 nM
Y11	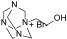	FAK-FERM inhibitor	Breast, colon cancer	50 nM
Y15	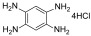	FAK-FERM Inhibitor	Breast, colon, thyroid, pancreatic cancer	1 μM
PF-573228	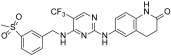	ATP-competitive inhibitor	Breast, lung cancer	4 nM
PF-43196	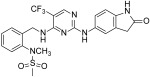	ATP-competitive inhibitor	Haematological cancers	2 nM

**Table 3 ijms-23-06381-t003:** Chemical scaffold and references of most recent compounds investigated as FAKIs.

Chemical Scaffold	Type of Inhibitors	Compounds	Ref.
diphenylpyrimidine	Reversible ATP-competitive inhibitor	1–12	[24,25,26,27,28,29,30,31,32,33,34,35,36,37]
diphenylpyrimidine	Irreversible ATP-competitive inhibitor	13–16	[38,39,40,41,42]
triazine	Reversible ATP-competitive inhibitor	17	[43,44,45,46]
Pirrolo-pyrimidine	Reversible ATP-competitive inhibitor	18–21	[47,48,49,50,51,52,53]
Thieno-pyrimidine	Reversible ATP-competitive inhibitor	22,23	[54,55]
1,3,4-oxadiazole	Reversible ATP-competitive inhibitor	24,25	[56,57,58]
1,2,4-triazole	Dual HDAC/FAKIs	26,27	[59,60]
benzotriazole	FAK/Pyk2/chelating inhibitors	28	[61]
quinoline	Dual EFGR/FAKIs	29	[62,63]
thiosemicarbazone	Not reported	30	[64]

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
