# Peer review of "The Development of FAK Inhibitors: A Five-Year Update"

_ijms, 2022, doi:10.3390/ijms23126381_

Round 1
Reviewer 1 Report
In this review, authors summarized the development of FAK inhibitors with emphasis on anticancer activity. They reviewed the biological and structural features of FAK inhibitors, as well as updated the pre-clinical studies and clinical uses of FAK inhibitors. Generally, this review is well organized and properly prepared with complete introduction and interpretation of most FAK inhibitors for cancer treatments. Several suggestions are provided for improvement of the present review.
1. Subhead 3.3.2 is missing and the number of subheads should be reorganized.
2. The molecular and structural characteristics of the compounds with FAK inhibitory activity are well addressed however, the relative or associated biological pathways involved in anticancer activities appear insufficient.
3. As mentioned in P.15, PTK2-PROTAC based on VHL and cereblon ligands using BI-3663 and BI-0319 scaffold did not show inhibitory effect on cell proliferation of HCC cells. Authors may further discuss why these compounds have good action on PTK2 but no influence on HCC cells.
Additionally, there are still grammar and format errors, authors should properly correct them.
Author Response
please see the attachement

Reviewer 2 Report
In this paper dr. Bruni and co-workers describe current state of investigation on Focal Adhesion Kinase (FAK) inhibitors. The review covers the research reports since 2017 and describes the latest achievements in this subject. FAK is non-receptor protein tyrosine playing distinct roles in cells. This protein promotes cells proliferation, focal adhesion, and migration. FAK mediated signalling induce resistance to apoptosis resulting from various stimuli, including oxidative stress, ultraviolet (UV) irradiation, exposure to anticancer drugs and others. The present work is important due to a fact that FAK is currently consider as a target in anti-tumour therapy. The content of the manuscript is comprehensive and well organise. The introduction provides a reader in a scope of FAK properties and importance in cancer development. The discussed inhibitors were classified to main groups of derivatives and presented in a form of tables containing structures and inhibitory activity (IC50). In several cases, available docking data for explanation of observed inhibitory activity were used. Authors discuss the structure-inhibitory activity relationship for series of derivatives. The references are up to date. The submitted manuscript is suitable for publication in present form.
Author Response
please see the attachement
